# Lexical Repetitions Lead to Rote Learning: Unveiling the Impact of Lexical Overlap in Train and Test Reference Summaries

**Prafulla Kumar Choubey**    **Alexander R. Fabbri**    **Caiming Xiong**    **Chien-Sheng Wu**
Salesforce AI Research
{pchoubey, afabbri, cxiong, wu.jason}@salesforce.com

## Abstract

Ideal summarization models should generalize to novel summary-worthy content without remembering reference training summaries by rote. However, a single average performance score on the entire test set is inadequate in determining such model competencies. We propose a fine-grained evaluation protocol by partitioning a test set based on the lexical similarity of reference test summaries with training summaries. We observe up to a 5x (1.2x) difference in ROUGE-2 (entity recall) scores between the subsets with the lowest and highest similarity. Next, we show that such training repetitions also make a model vulnerable to rote learning, reproducing data artifacts such as factual errors, especially when reference test summaries are lexically close to training summaries. Consequently, we propose to limit lexical repetitions in training summaries during both supervised fine-tuning and likelihood calibration stages to improve the performance on novel test cases while retaining average performance. Our automatic and human evaluations on novel test subsets and recent news articles show that limiting lexical repetitions in training summaries can prevent rote learning and improve generalization.

## 1 Introduction

Modern neural summarization models generate summaries that obtain high evaluation scores on automated metrics such as ROUGE. While models obtain high scores on average, factors beyond their summarization-related capabilities, such as learning of data artifacts, may contribute to performance. For instance, we show summaries generated by the *BART* (Lewis et al., 2020), *BRIO* (Liu et al., 2022) and *PEGASUS* (Zhang et al., 2020) models trained on XSUM data (Narayan et al., 2018) for a recent news article in Table 1. They all contain the same factual error "*investigation on Russia's meddling in 2016 election*," which is also mentioned several times in reference XSUM summaries. Had

| |
|---|
| *Article:* The American people may soon hear directly from a key witness in the House **January 6 select committee's investigation** who can speak to former President Donald Trump's approving reaction to the **US Capitol riot** – live testimony that could be the most important moment... |
| *BART:* A key witness in the House Intelligence Committee's Russia investigation is likely to testify in person, according to a source. |
| *BRIO:* The special counsel's investigation into Russia's meddling in the 2016 election and the Trump campaign's links to it is expected to return to the public spotlight this week. |
| *PEGASUS:* A key witness in the House Intelligence Committee's investigation into Russian interference in the 2016 election has changed her lawyer, sources have told CNN. |

Table 1: All *BART*, *BRIO* and *PEGASUS* models trained on XSUM data incorrectly generates *investigation on Russia's meddling in 2016 election* instead of *January 6 capitol attack investigation*.

the document been taken from XSUM, the fact *investigation on Russia's meddling* would likely be correct due to the temporal adjacency among facts in reference testing and training summaries. However, this is an undesirable behavior from a summarization model given it ignores the true fact "*investigation of January 6 attack*" in the source document.

Since most summarization datasets are created by randomly partitioning data into train, validation, and test sets, training and test summaries also contain similar distributions of summary-worthy information. Therefore, we argue that the existing practice of exclusively reporting average performance on the full test set may overestimate the true capability of a model to summarize the most salient information from a document describing novel information that is unseen in training documents. Similar to the example in Table 1, a model may mysteriously generate the information it has seen in reference training summaries but not the most salient or correct information according to the given document.

To validate this, we divide test data into subsets with different levels of lexical similarity relative to reference training summaries. We use the percentage of 4-grams in a reference test summary that overlaps with the 4-grams in training summaries. We assume low lexical similarity between test and training reference summaries likely represents novel summary-worthy information in the corresponding test document. We, then, evaluate the current SoTA summarization models, *BART*, *PEGASUS*, *BRIO* and *REINA* (Wang et al., 2022), on different test subsets. Indeed, we find that all models systematically underperform on the test subset with low lexical overlap with training summaries. Our fine-grained analyses (§4) further show that higher lexical similarity between training and test reference summaries may also result in higher remembered facts in generated summaries.

Interestingly, since all models generate the same factual error in Table 1, it is evident that the above behavior results from data artifacts, the repeated mention of event *'investigation on Russia's meddling'* in XSUM training summaries. Therefore, we perform a systematic study (§5.2) to analyze the effect of lexical repetitions in training summaries. We show that increasing training data size by adding training samples with summaries similar to other training summaries helps in improving the average performance of a summarization model, but not on test sets with low lexical overlap with training summaries. At the same time, it also leads to more hallucinated errors when summarizing new documents due to remembering by rote.

Naively, we can improve the performance on documents with novel summary-worthy content or reduce remembered factual errors by increasing the diversity of training summaries, as demonstrated in our empirical results in § 5.2. We find that models trained on a small yet diverse subset of the training samples have fewer entity remembering issues. However, this requires removing a large proportion of training samples and sacrificing the model's performance on those summaries that are similar to reference training summaries.

We aim to use train data such that the resulting summarization model performs well on novel test summaries while retaining most or all of its average performance. Accordingly, we propose to first fine-tune a base summarization model on all training data, and then calibrate (Liu et al., 2022; Zhao et al., 2022) that model on a subset of training data that maximizes lexical diversity. This first stage of fine-tuning enables the models to assign high probabilities to reference summaries, reaping the maximum benefit from all training samples. For the second-stage calibration, we use the *BRIO*-training paradigm, which teaches the base model to rank generated summaries by ROUGE score, on a subset of the most diverse training samples. Thus, we guide the model to only promote the rank of its generated summaries with higher lexical diversity and minimize repetitions across different summaries. Through empirical studies in this paper, we show

- systematic patterns between the performance of a summarization model and lexical similarity of testing and training summaries.

- lexical repetitions in training summaries may over-fit a model, and make it remember and generate facts from training summaries.

- that controlling lexical repetition in training summaries during fine-tuning or calibration can prevent rote learning and improve generalization.

## 2   Related Work

Several summarization works (See et al., 2017; Fu et al., 2020; Nair and Singh, 2021) have studied repeated ngrams within a generated summary and proposed various means to minimize that. Recently, Salkar et al. (2022) shows the tendency of summarization models to self-repeat, i.e., repeat certain ngrams across different summaries. However, these works exclusively focus on minimizing or quantifying ngram repetitions within or across generated summaries. On the other hand, we use ngram overlaps to identify novel reference summaries in a dataset and focus on complementary analyses, such as, how the average performance of a model correlates with the lexical similarity between reference training and testing summaries.

Among the works that focus on artifacts in summarization data, Nan et al. (2021) excludes noisy summaries from training data, Kano et al. (2021) defines training curriculum based on the level of noise in a training sample, Choubey et al. (2021) uses contrastive ensembling between models trained on clean and noisy summaries, and Adams et al. (2022) revises noisy training summaries through an auxiliary model. They all focus on factual errors in generated summaries and study the correlation between model hallucination

and training data noise. To the best of our knowledge, we are the first to analyze the correlation between the lexical diversity of training reference summaries and information correctness and relevance of generated summaries and use that to improve the summarization model's performance.

## 3 Datasets and Evaluation Setup

In our experiments, we consider four summarization datasets, CNN (Hermann et al., 2015), XSUM, WikiHow (Koupaee and Wang, 2018) and SamSum (Gliwa et al., 2019). CNN consists of highly extractive article highlight summaries on CNN and Daily Mail news articles. XSUM includes highly abstractive, one-sentence summaries of BBC news articles. WikiHow contains articles from the WikiHow[1] knowledge base, with summaries that are concatenation of paragraph outline sentences. SamSum includes chat dialogues with human-written reference summaries.

The four datasets differ in content organization structure and domain. XSUM and CNN summaries are lead-biased following the inverted pyramid writing style of news articles. Contrarily, source articles in WikiHow and SamSum datasets contain summary-worthy content throughout the text. Secondly, XSUM, CNN and WikiHow articles are structured text written in the third-person point of view while SamSum involves multiple participants talking about themselves. These variations help evaluate the generic effects of lexical repetitions in summarization models and datasets.

**Test Data Partitioning using Lexical Overlap with Training Summaries:** We use 4-gram overlap to partition each test dataset into subgroups with different levels of lexical similarity (or novelty) with respect to training summaries. We choose the n-gram size of four to ensure sufficient samples in all partitions while estimating similarity. We first make a list of all unique 4-grams in the training summaries. Then, for a test summary, we calculate the percentage of its 4-grams that also exist in the training 4-grams list. Finally, we rank the test samples based on their percent 4-gram overlap and partition them into disjoint subsets such that each subset contains a sufficient and comparable number of samples. We fix the minimum percent overlap range width for each subset to 5 and increase the width in a multiple of 5 to adjust the

[1] https://www.wikihow.com

number of samples in a subset. We report statistics for each test data partition in the Appendix (Tables 4, 5, 6 and 7). We call the test data subset with the smallest (largest) percent 4-gram overlap as $T_{nov}$ ($T_{sim}$).

## 4 Fine-grained Evaluation of SoTA Summarization Models

We perform fine-grained evaluations for *BART*, *PEGASUS* and *BRIO* models on XSUM and CNN, and additionally *REINA* model on XSUM. *BRIO* uses a summarization model (*BART* or *PEGASUS*) to generate multiple candidate summaries with different levels of quality and then encourages the model to assign higher probabilities to better candidates using a contrastive loss. *REINA* retrieves a training document most similar to the given test document. It then concatenates the retrieved training document summary with the test document during inference. We summarize our findings in Figures 1 and 2 below:

_Lexical overlap between test and train summaries drives the higher average test performance._ We use ROUGE-2 (R2) and Entity Recall ($E_{Rec}$). $E_{Rec}$ measures the percentage of entities in a generated summary that are present in both its reference summary and its source article. Entities can be used as a proxy to measure the information relevance of a summary (Zheng et al., 2020). As shown in Fig. 1a and 1d, all models obtain significantly lower R2 scores on the most novel test subset ($T_{nov}$) for both XSUM and CNN. As the expected reference summary gets closer to training summaries, R2 for all models increases. Additionally, the relative difference in performance between $T_{sim}$ and $T_{nov}$ is much higher for the XSUM data (up to 5x vs 1.5x for the CNN data). This is explainable by the different characteristics of the two datasets, higher abstractivenes and data artifacts in XSUM make it difficult for a model to generate summaries when expected summary-worthy contents are different from the training summaries. All models also follow a similar pattern on $E_{Rec}$ for the CNN data (Fig. 1e), performing much worse on $T_{nov}$. For instance, the *BRIO* model has an absolute $E_{Rec}$ gap of over 10% between $T_{sim}$ and $T_{nov}$. However, on XSUM, we observe that $E_{Rec}$ for all models initially increases and then starts to drop when the percent lexical overlap increases above 35% (Fig. 1b). This result stems from the data artifact discussed next.

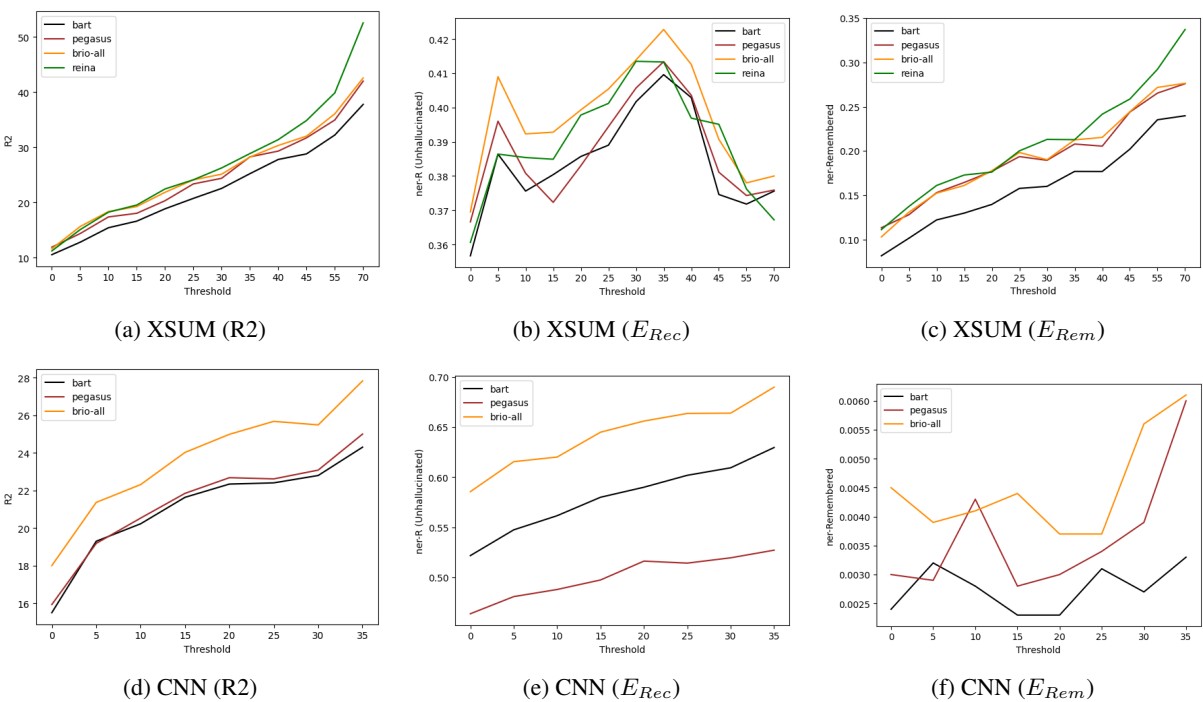

(a) XSUM (R2)  (b) XSUM ($E_{Rec}$)  (c) XSUM ($E_{Rem}$)

(d) CNN (R2)  (e) CNN ($E_{Rec}$)  (f) CNN ($E_{Rem}$)

Figure 1: R2, $E_{Rec}$ and $E_{Rem}$ scores comparison of recent neural models on different CNN and XSUM test data partitions.

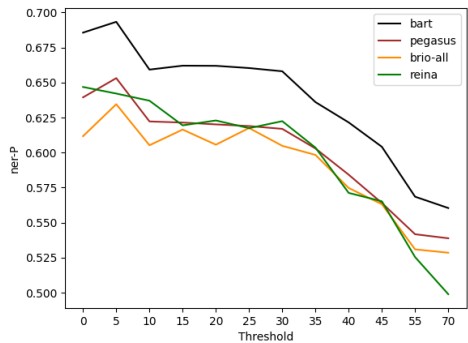

Figure 2: $E_{Prc}$ score comparison of recent neural models on different XSUM test data partitions.

*Models may generate factual errors when the lexical overlap between test and training summaries increases.* We plot remembered entity ($E_{Rem}$, XSUM: Fig. 1c and CNN: Fig. 1f) and entity precision ($E_{Prc}$, XSUM: Fig. 2). $E_{Rem}$ measures the percentage of entities in a generated summary that are present in its reference summary but not in its source article. $E_{Prc}$ measures the percentage of entities in a generated summary that are also present in its source article. We exclude $E_{Prc}$ plot for the CNN dataset since it is highly extractive and copies entities from the document as evident from extremely low $E_{Rem}$ (∼0.6%).

As train-test reference summary lexical over-

lap increases, $E_{Rem}$ increases and $E_{Prc}$ decreases for all models. For instance, from $T_{nov}$ to $T_{sim}$, the percentage of remembered entities for *REINA* increases more than 3x on the XSUM data. Remembered entities are akin to factual hallucinations (such as information that might be correct due to relatedness of facts in training and test reference summaries, e.g., *investigation on Russia's meddling in 2016 election* for a document from XSUM test set, Fig. 1) and as expected they are prevalent in test cases with reference summaries having a high percent train 4-gram overlap. It is worth noting that all models obtain higher entity precision (or, generally, reproduce fewer extrinsic entity errors) on the test subsets with the lowest 4-gram overlap with training summaries, indicating that models would likely be factually more consistent if the expected summaries were lexically dissimilar to the training summaries.

We also perform fine-grained evaluations for *BART* and *BRIO* models on WikiHow and SamSum datasets, as shown in Fig. 6 in Appendix, observing similar trends. Higher lexical overlap between the train and test reference summaries leads to higher R2, $E_{Rec}$ and $E_{Rem}$ scores. On SamSum, however, $E_{Rem}$ decreases slightly with the increase in the lexical overlap. This may not be surprising given the smaller size of the SamSum

dataset (10-20x smaller than other datasets) and the very low lexical overlap between the training and test reference summaries.

**Implications for Modelling:** We found that all models perform consistently worst on the $T_{nov}$. Analyzing *BRIO* and *REINA* on XSUM, neither model improves R2 over *PEGASUS* on the $T_{nov}$. While *BRIO* uniformly improves in R2 over *PEGASUS* on the remaining XSUM test subsets other than the $T_{nov}$, *REINA* obtains relatively higher R2 gain on test subsets with very high train-lexical overlap. Similarly, on $E_{Rem}$, *REINA* and *BRIO* models perform comparably on test subsets with lower train-lexical overlap, but as the train-lexical overlap increases, *REINA* generates more remembered entities. This is not surprising since *REINA* retrieves and uses training summary as context. Such retrieval-based models are likely to bias the generated summaries toward the retrieved training summary.

Our fine-grained analysis, thus, provides deeper insight into a model's capabilities, e.g., a higher average performance for *REINA* on XSUM results from improvement on test subsets that are similar to training summaries, and it may obtain higher performance scores only if the reference summaries are expected to be lexically similar and containing the same factual information (or hallucination) as the training summaries.

# 5 Effect of Lexical Repetitions in Reference Training Summaries

In section §4, we show that high lexical overlap between the training and test reference summaries drives high average performance for all neural summarization models. The lower R2 and $E_{Rec}$ on $T_{nov}$ or higher $E_{Rem}$ and low $E_{Prc}$ on $T_{sim}$ suggest that all models are somewhat memorizing and generating information in reference training summaries and learning the dataset or its artifacts besides learning the task-related summarization capabilities. Given these findings, what factors are driving the high-performance gap between $T_{nov}$ and $T_{sim}$? Intuitively, the above behavior of models could be a result of lower information diversity in training data. To validate our hypothesis and study model behavior relative to training data diversity, we evaluate the effects of training 4-grams repetitions (§5.1) in both model fine-tuning (§5.2) and likelihood calibration (§5.3). We also evaluate the effects of 4-grams repetitions in zero-shot

settings, described in Appendix §B.1.

## 5.1 Controlling Lexical Repetitions in Training Summaries

Given a training dataset $D$, our goal is to select a subset ($D_{\theta_{4G}}$) such that no 4-gram in $D_{\theta_{4G}}$ is repeated more than a predefined threshold ($\theta_{4G}$). We initialize $D_{\theta_{4G}}$ with an empty set and track the number of repetitions of all 4-grams ($4G$) in $D_{\theta_{4G}}$. While iterating through each sample in $D$, we check if including the current sample increases the frequency of any 4-gram in $4G$ above the threshold $\theta_{4G}$. If it does, we exclude the current sample. Otherwise, we include the current sample in $D_{\theta_{4G}}$ and accordingly update the 4-grams repetitions count in $4G$. In all our experiments, we first randomize the order of training samples and then build three training data subsets (different random seeds) for each threshold. We train three models for each $\theta_{4G}$ and report the average performance.

## 5.2 Supervised Fine-tuning

We control the lexical diversity of the actual training dataset following the steps described in §5.1. We choose different $\theta_{4G}$ for each dataset to ensure: 1) sufficient training samples for the lowest threshold, and 2) a proportionate increase in the number of training samples from the lower to the higher $\theta_{4G}$. We plot the performance of models trained on XSUM, CNN, WikiHow and SamSum datasets with different thresholds ($\theta_{4G}$) in Fig. 3 and 4. In each plot, we normalize the performance with respect to the performance of the model trained with the lowest $\theta_{4G}$. The actual $\theta_{4G}$ and the corresponding number of samples for each dataset are shown in the Appendix (Table 8).

We observe that increasing training data size by adding samples with lexically similar summaries:

*Improves R2/ $E_{Rec}$ on test summaries with similar lexical distributions:* Models trained with the lowest $\theta_{4G}$ obtain R2 (up to 95%) comparable to the models trained on all data on $T_{nov}$ (Figs. 3a, 3d, 3g and 3j). Additional samples, with repeated 4-grams, mainly help with the test summaries having overlapping 4-grams with the training summaries. Results on $E_{Rec}$ (Figs. 3b, 3e and 3h) follow the pattern similar to the R2 metric on XSUM, CNN and WikiHow datasets. But the model trained on all SamSum data obtains the best entity recall on the $T_{nov}$ subset (Fig. 3k), that may again not be surprising given the SamSum dataset has sparse

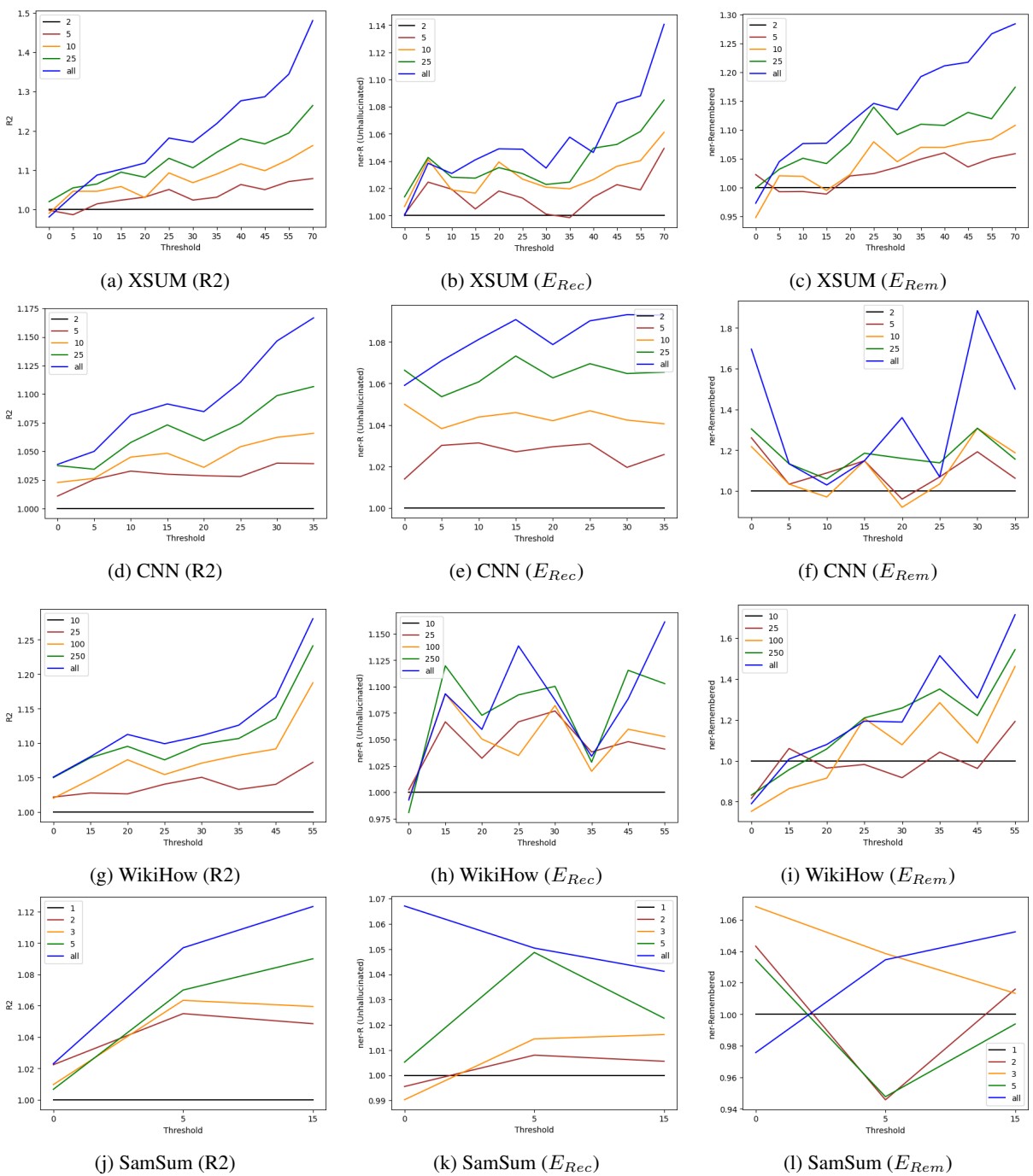

Figure 3: R2, $E_{Rec}$ and $E_{Rem}$ scores comparison of *BART* models, trained with different n-grams repetition thresholds ($\theta_{4G}$), on different test data partitions.

4-gram repetitions ($\theta_{4G}$ of 5 covers >75% of the training set) with only 14.7K training samples.

Unsurprisingly, the relative improvement on a test subset is also proportional to its lexical similarity with the training summaries, with the maximum improvement seen on the $T_{sim}$, which aligns with our observations in §4.

*Makes models susceptible to remembering facts instead of inferring them from the source articles:*

We plot $E_{Rem}$ for models trained for different $\theta_{4G}$ in Fig. 3c, 3f, 3i and 3l. Additionally, since summarization models trained on XSUM remember and generate facts from training summaries (e.g., *Russia's meddling investigation* in Fig. 1), we also plot $E_{Prc}$ (Fig. 4) for different models on the XSUM data. As expected, repeated 4-grams in training summaries (i.e., higher $\theta_{4G}$) lead to more hallucinated facts (high remembered entities) and more

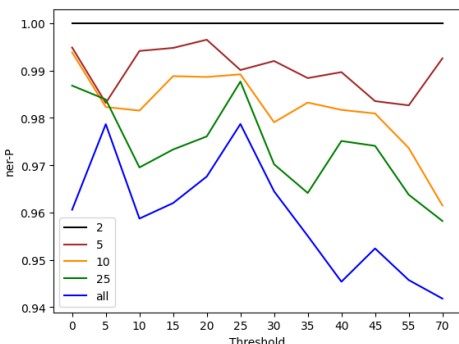

Figure 4: $E_{Prc}$ score comparison of *BART* models, trained with different n-grams repetition thresholds ($\theta_{4G}$), on different XSUM test partitions.

|  | XSUM | CNN | WikiHow | SamSum |
|---|---|---|---|---|
| *BRIO*-2/10/1 | 24.85 | 23.39 | 18.60 | 27.31 |
| *BRIO*-5/25/2 | 24.90 | 23.50 | 18.75 | 27.64 |
| *BRIO*-10/100/3 | 24.94 | 23.54 | 19.12 | 27.64 |
| *BRIO*-25/250/5 | 24.99 | **23.79** | 19.16 | 27.72 |
| *BRIO*-all | **25.31** | 23.73 | **19.34** | **27.90** |

Table 2: Average performance of *BRIO* models calibrated using training data subsets with different $\theta_{4G}$. Models are named following the notation *BRIO*-{$\theta_{4G}$ for XSUM and CNN/ WikiHow/ SamSum}.

factual errors (low entity precision). Further, increasing $\theta_{4G}$ keeps reducing $E_{Prc}$ on $T_{nov}$ without increasing $E_{Rem}$, highlighting that rote learning makes model remember facts from training summaries which are not related to the information in a given document and would harm the factual accuracy of models. In the case of SamSum, higher 4-gram overlap does not follow the same pattern for $E_{Rem}$ as observed in other datasets, which is consistent with previous findings on SamSum.

Factual hallucinations (e.g., higher $E_{Rem}$ on $T_{sim}$) may seem correct due to similar facts repeated in both training and test reference summaries. It is not immediately clear if such factual hallucinations are a strength or weakness of a model. Therefore, we perform intervention-based analyses by selecting an entity from a reference test summary and editing the corresponding test document with respect to the selected entity, and then evaluating the factual accuracy of the newly generated summary based on the intervened document. We observe that a model trained on data with higher lexical repetitions remembers entities in training data and is also more likely to generate them, overlooking the interventions on the document. Details of the evaluation and result are discussed in Appendix C.

Secondly, in the above experiments, we analyze whether including new lexically similar training summaries improves model performance, which increases lexical repetitions by adding new training samples. However, a fairer comparison would require maintaining the size of the training data. Therefore, we also evaluate models trained on a comparable number of samples that have low 4-gram repetitions and high (or randomly chosen) 4-gram repetitions. We discuss the experimental

details in Appendix B.2. We find that higher 4-gram repetitions in training data give significantly higher performance on $T_{sim}$ and also the best average performance even when the training data size is fixed. But, fewer training 4-gram repetitions result in better generalization as indicated by its superior performance on $T_{nov}$. Evidently, preventing 4-gram repetitions yields the model with the best generalization performance.

### 5.3 Likelihood Calibration

In §5.2, we show that controlling ngram repetitions in training summaries can prevent rote learning during fine-tuning. However, that results from excessive repetitions of related factual errors in XSUM training summaries. When reference training summaries generally contain correct factual information (e.g., CNN data), we want to use all the available training data and maximize information recall on test summaries that are similar to training summaries. At the same time, we would also like to improve the performance of a model on the documents with novel summary-worthy content.

In such a case, where data or a model suffers less from factual errors, we first use all training samples to train the base summarization model. This enables a model to assign high probabilities to information in reference training summaries. Then, we use different training data subsets with different $\theta_{4G}$ to calibrate the base model, following the *BRIO*-training loss proposed by Liu et al. (2022). *BRIO* learns to assign high probability mass to a generated summary if it obtains a higher ROUGE score compared to the reference training summary. So, by controlling the diversity of training summaries, we can teach the model to assign relatively higher probabilities to diverse generated summaries. The resulting comparisons for models on different XSUM/CNN test subsets are shown in Fig. 5. We also report the average R2 scores for different *BRIO* models in Table 2. We name models following the notation *BRIO*-$\theta_{4G}$.

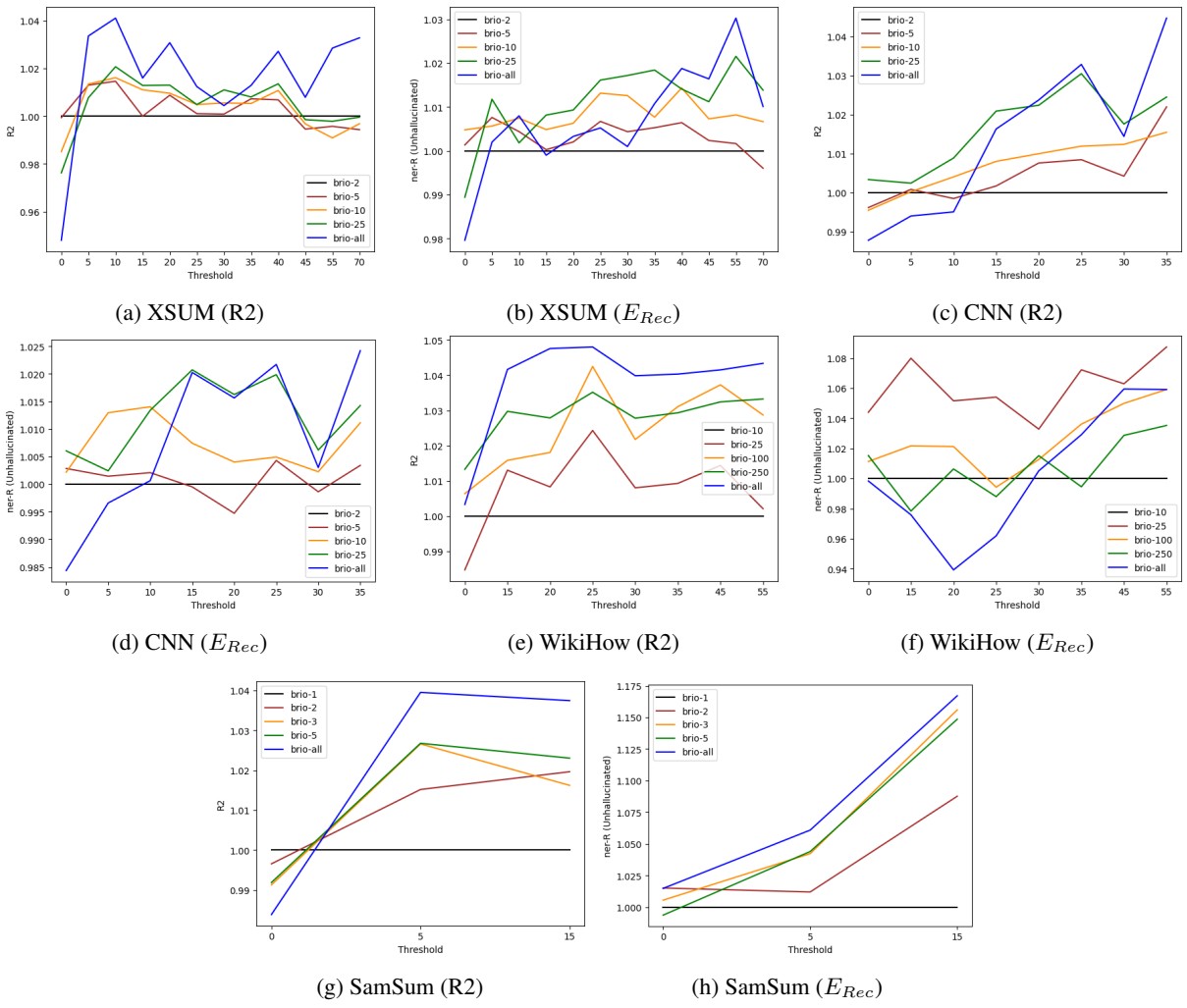

Figure 5: Performance comparison of *BRIO* models, calibrated with different n-grams repetition thresholds, on different test data partitions. Models are name following the notation *BRIO-$\theta_{4G}$*.

Similar to fine-tuning, we observe that *BRIO-all* obtains lower R2 or $E_{Rec}$ than a model trained with a data subset with fewer 4-gram repetitions on the $T_{nov}$. Interestingly for CNN, 1) *BRIO-25* performs comparably to *BRIO-all* on average R2 (Table 2), while performing better on the $T_{nov}$, 2) *BRIO-25* obtains higher $E_{Rec}$ both on average and on the $T_{nov}$ subset (*BRIO-25*: 64.65 vs *BRIO-all*: 63.49 on average $E_{Rec}$). For XSUM (WikiHow), *BRIO-25* obtains higher $E_{Rec}$ for the majority (all) of test subsets. $E_{Rec}$ result on the SamSum dataset is again an exception with *BRIO-all* obtaining the best performance, which is identical to the behavior for supervised training.

In summary, models calibrated on all data always perform the best on $T_{sim}$ on recall-oriented metrics, but they perform badly on $T_{nov}$ and they may not always result in the best average performance.

## 6 Discussion: The Performance Trade-off and Human Evaluation

From our experiments in § 5.2 and § 5.3, we observe that using all training samples is the most plausible choice for obtaining the best average performance. However, the improvement is not uniform across all test subsets. Further, the intervention-based analysis on XSUM shows that higher ngram repetitions make a model learn and reproduce data artifacts. To determine what makes the best strategy for utilizing the available training data, we perform a human study to compare the generic summarization capabilities of different models.

We use pairwise evaluation between supervised models fine-tuned on $\theta_{4G}$ of 25 and all data for each of the XSUM and CNN datasets. We also compare *BRIO-25* and *BRIO-all* models trained on each of

|        | 25 (FT) | all (FT) | *BRIO-25* | *BRIO-all* |
|--------|---------|----------|-----------|------------|
| Consistency | | | | |
| XSUM   | 17%     | 13%      | 19%       | 7%         |
| CNN    | 7%      | 2%       | 9%        | 8%         |
| Relevance | | | | |
| XSUM   | 34%     | 24%      | 36%       | 13%        |
| CNN    | 30%     | 29%      | 34%       | 28%        |

Table 3: Percentage of times human annotators labeled a summarization system better than the other.

the XSUM and CNN datasets. We choose $\theta_{4G}$ of 25 for comparison since the resulting models perform the best (or comparable to the model trained on all data) on $T_{nov}$ for both XSUM and CNN for both fine-tuning and calibration.

We conduct our study on 100 recent CNN articles from Goyal et al. (2022) to maximize temporal exclusivity of information (events and entities) and minimize the influence of train-lexical overlap on relevance and correctness of generated summaries. This also mimics the practical use-case scenario where there is a temporal separation between the training and test data. Since the 100 articles do not have reference summaries, we first ask our annotators (three authors) to highlight the most relevant information for each document, analogous to prior reference-free human evaluation (Hardy et al., 2019).

We compare the number of document highlights that are present in summaries generated by the two systems. If both systems include the same number of highlights, both systems are labeled equivalent on relevance. Otherwise, the system that generated more highlighted information is rated better on relevance. For consistency, we compare the number of hallucinated facts in summaries generated by two systems and rank the system with fewer hallucinations as better. The inter-annotator agreements for relevance and consistency are 0.64 and 0.82 (Krippendorff, 2011), respectively. We report the percentage of documents for which each model is rated better than the other in Table 3.

First, we observe that models fine-tuned on $D_{\theta_{4G}=25}$ generate more consistent summaries compared to the models that use all training data. This corroborates our results based on automated entity precision metric ($E_{Prc}$) as well as intervention-based analysis on XSUM. On the relevance scale, we can observe that XSUM-25 (FT) generates more relevant summaries compared to the XSUM-all (FT), suggesting that controlling lexical repetitions in training data with artifacts is a superior technique

to improve model performance. Meanwhile, CNN models trained on $D_{\theta_{4G}=25}$ or all data achieve similar results. The main reason could be that the CNN dataset has fewer artifacts.

In addition, comparing the results of *BRIO-25* and *BRIO-all* on both CNN and XSUM datasets, we see that controlling lexical repetitions during calibration is an effective strategy. This is also reflected in the results in § 5.3. Another advantage to conducting the calibration step with fewer but lexically diverse training samples is that it is computationally more expensive than the supervised fine-tuning (Liu et al., 2022). Thus, our proposed solution can achieve uniform improvement across all test subsets and also minimize the computational cost.

# 7 Conclusion

In this work, we propose a fine-grained evaluation protocol by partitioning a test set based on the lexical similarity between reference test summaries and training summaries to evaluate the generalization capabilities of summarization models. Our comparative evaluations of current SoTA summarization models show that they all systematically underperform in generating novel summary-worthy content. Next, we show that higher lexical repetitions in training data contribute to this phenomenon. In addition, training repetitions make models vulnerable to learning data artifacts. Finally, through a series of automatic and human evaluations, we show that controlling the lexical diversity of training data at different stages of training can help a model generate more relevant and consistent summaries on novel test subsets.

## Limitations

The datasets utilized in this research contain documents and summaries in English and thus mainly represent the culture of the English-speaking populace. Gender, age, political or other biases may also exist in the dataset, and models trained on these datasets may propagate these biases.

Our experiments and analyses are based on the assumption that training data contains artifacts. Also, it is evident from the results, that the effectiveness of our proposed models is best for the noisiest XSUM dataset. So, our analytical results and improvement from a model may have limited implications on a perfect dataset that does not exhibit any learnable artifacts.

We relied on automated metrics, such as entity recall for measuring information relevance, entity precision for information correctness, and remembered entity for factual hallucinations in all our experiments. These metrics use lexical matching on automatically extracted entity mentions and are error-prone. Exclusively for a subset of models, that perform the best according to automated metrics, we use human annotations for additional evaluations.

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

## A  Experimental Details

We use the Huggingface Transformers library (Wolf et al., 2020) (PyTorch (Paszke et al., 2017)) for all our experiments. We use batch size of 64 for the XSUM, CNN and WikiHow datasets, and batch size of 32 for the SamSum dataset. We start with the *"facebook/bart-large"* checkpoint and fine-tune it for 5 (3) epochs on XSUM, CNN and WikiHow (SamSum) datasets using default training hyper-parameters (optimizer: Adam, learning rate: 5e-5, $\beta_1$: 0.9, $\beta_2$: 0.999, $\epsilon$: 1e-8). Similarly, we use default inference parameters for each dataset and model. All training and evaluations are performed using 40 GB Nvidia A100 GPUs.

We use NLTK (Bird et al., 2009) for word tokenization and ngrams generation, and Spacy (*en_core_web_sm*) (Honnibal et al., 2020) for extracting entities and use token-level overlap for calculating $E_{Rec}$, $E_{Prc}$ and $E_{Rem}$ scores. Since a model may remember or hallucinate any type of entity, we use all entity types (excluding stop words) for calculating these metrics. We use the HuggingFace Datasets library (Lhoest et al., 2021) for calculating ROUGE scores (Lin, 2004).

| XSUM | 0-5 | 5-10 | 10-15 | 15-20 | 20-25 | 25-30 | 30-35 | 35-40 | 40-45 | 45-55 | 55-70 | >70 |
|---|---|---|---|---|---|---|---|---|---|---|---|---|
| #Samples | 690 | 826 | 961 | 1048 | 1159 | 1216 | 1029 | 863 | 888 | 1152 | 681 | 819 |

Table 4: Percent overlap range and number of samples in the XSUM test data partitions.

| CNN | 0-5 | 5-10 | 10-15 | 15-20 | 20-25 | 25-30 | 30-35 | >35 |
|---|---|---|---|---|---|---|---|---|
| #Samples | 789 | 1587 | 2118 | 2039 | 1726 | 1350 | 823 | 1058 |

Table 5: Percent overlap range and number of samples in the CNN test data partitions.

| WikiHow | 0-15 | 15-20 | 20-25 | 25-30 | 30-35 | 35-45 | 45-55 | >55 |
|---|---|---|---|---|---|---|---|---|
| #Samples | 640 | 445 | 601 | 702 | 628 | 1120 | 817 | 1043 |

Table 6: Percent overlap range and number of samples in the WikiHow test data partitions.

| SamSum | 0-5 | 5-15 | >15 |
|---|---|---|---|
| #Samples | 286 | 246 | 287 |

Table 7: Percent overlap range and number of samples in the SamSum test data partitions.

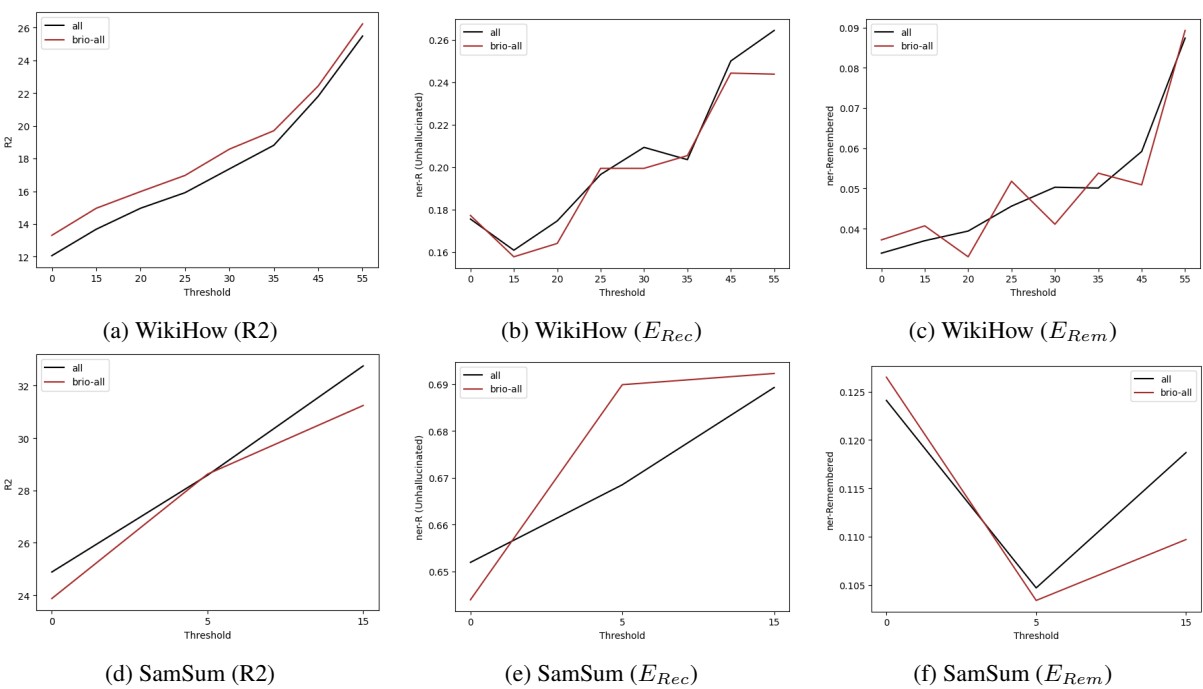

(a) WikiHow (R2)  (b) WikiHow ($E_{Rec}$)  (c) WikiHow ($E_{Rem}$)

(d) SamSum (R2)  (e) SamSum ($E_{Rec}$)  (f) SamSum ($E_{Rem}$)

Figure 6: Performance comparison of *BART* (all) and *BRIO* (*BRIO-all*) models (trained on all training samples) on different WikiHow and SamSum test data partitions.

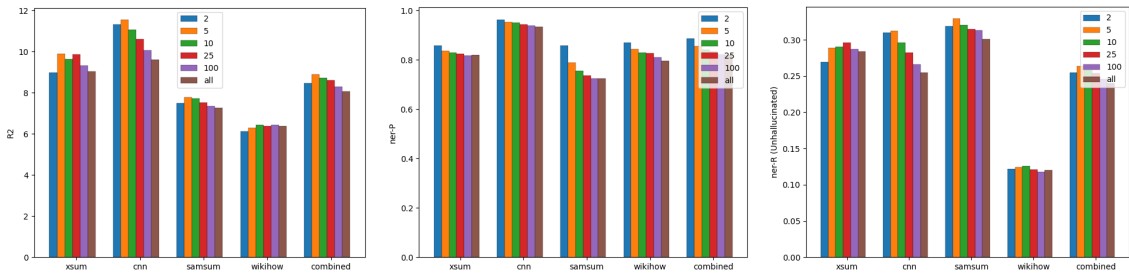

Figure 7: Zero-shot evaluation of models trained on Newsroom dataset.

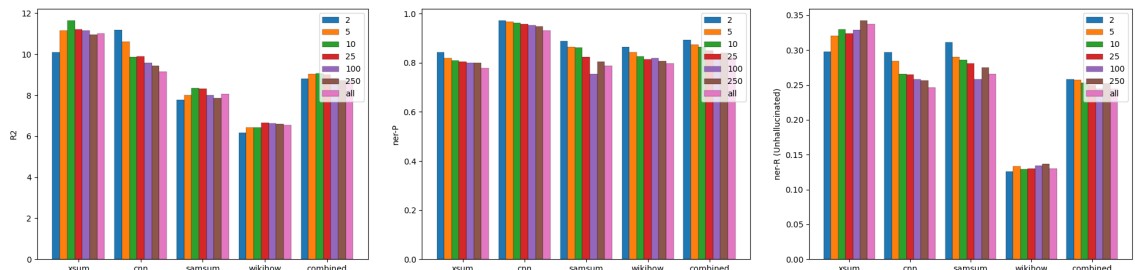

Figure 8: Zero-shot evaluation of models trained on C4 dataset.

| Threshold | 2 | 5 | 10 | 25 | all |
|---|---|---|---|---|---|
| XSUM | 44K | 76K | 102K | 136K | 204K |
| CNN | 44K | 86K | 124K | 177K | 287K |

| Threshold | 10 | 25 | 100 | 250 | all |
|---|---|---|---|---|---|
| WikiHow | 36K | 57K | 99K | 130K | 168K |

| Threshold | 1 | 2 | 3 | 5 | all |
|---|---|---|---|---|---|
| SamSum | 5.7K | 8.3K | 9.6K | 11.1K | 14.7K |

Table 8: Number of training samples for different 4-gram repetition thresholds ($\theta_{4G}$) for supervised training datasets.

|  | 2 | 5 | 10 | 25 | 100 | 250 |
|---|---|---|---|---|---|---|
| C4 | 249K | 510K | 823K | 1.48M | 3.26M | 5.06M |
| NR | 39K | 93K | 155K | 273K | 511K | - |

Table 9: Number of training samples for different 4-gram repetition thresholds for C4 (C4_newsalike) and NR (newsroom) datasets.

## B Effects of Lexical Repetition in Reference Training Summaries

### B.1 Zero-Shot Evaluation

For zero-shot evaluation, we follow the pre-training objective of *PEGASUS*. Given a document, we first calculate the ROUGE-1 (R1) score between each sentence and the rest of the document. Then, we select the sentence scoring the highest R1 score as the summary for the rest of the document. We experiment with two datasets, the training subset of newsroom dataset (Grusky et al., 2018) containing 988K documents and C4-newsalike (Raffel et al., 2022) dataset containing 13.64M documents. Using the strategy described in §5.1 for controlling lexical diversity, we create training data with different maximum 4-gram repetition thresholds and fine-tune the *BART* model. We train each model for one epoch since it is logical to train a model on more samples, even with repeated n-grams, to increase the number of training steps rather than training the model for more than one epoch on the same samples. We use the thresholds of 2, 5, 10, 25, and 100 (2, 5, 10, 25, 100, and 250) for newsroom (C4_newsalike) datasets. During inference, we follow the standard dataset-specific hyper-parameters. The sizes of training data subsets for each repetition threshold are shown in Table 9. We plot the R2, entity precision and entity recall results in Figs. 7 and 8.

We see similar effects of 4-gram repetitions for both *newsroom* and *C4_newsalike* training datasets despite having a 15x difference in their sizes. On all four evaluation datasets, both R2 and $E_{Rec}$ peaks at some middle $\theta_{4G}$ i.e. the performance increases until a certain $\theta_{4G}$ and then starts declining. The best $\theta_{4G}$ is different for each evaluation dataset. However, on combined evaluation datasets, the best thresholds for newsroom (C4_newsalike) are 5 and 5 (10 and 2) for R2 and entity recall respectively, which constitute less than ~10% (~6%) of newsroom (c4_newsalike) datasets. On entity precision metric, adding samples with repeated n-grams always decreases the performance. This is unsurprising given the pseudo-summaries are likely to contain extrinsic entity errors and repeated 4-grams make it easier for models to learn the dataset artifacts.

Summarizing our observations, **increasing data size by adding lexically similar summaries may harm both relevance and correctness of information in the zero-shot setting**. If we expect a summarization model to perform well zero-shot, then it would be logical to control lexical repetitions in training summaries.

### B.2 Supervised Fine-tuning: with Equal Number of Diverse and Random Samples

In § 5 and 6, we observe how adding more training samples with lexical repetitions influences the performance of a model, including benefits on test samples that are lexically similar to training summaries as well as the risk of learning data artifacts. However, we are constrained by the fact that in-

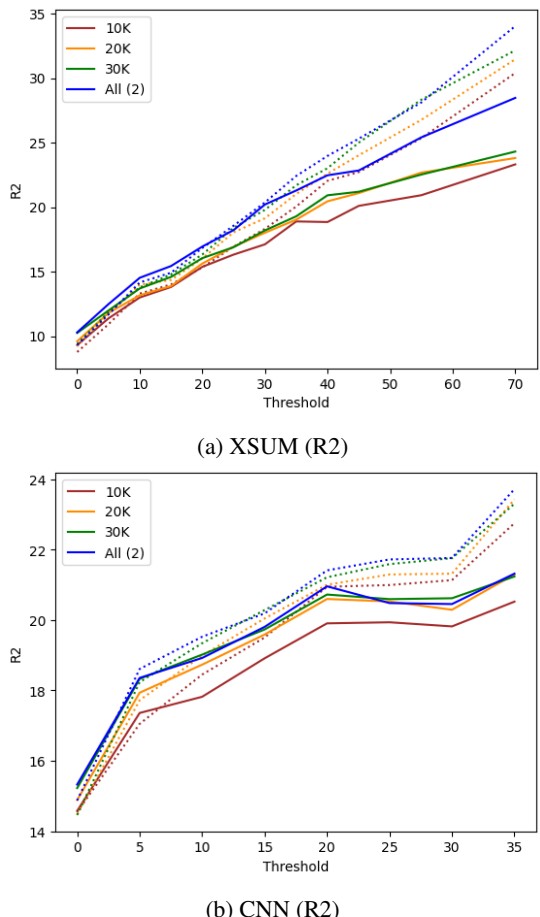

(a) XSUM (R2)

(b) CNN (R2)

Figure 9: R2 scores comparison of *BART* models trained on 10K, 20K, 20K and all (i.e. 44K) samples from diverse, $\theta_{4G}$ of 2 (solid), and random (dotted lines) training data on different test data partitions.

creasing lexical repetitions also increases the training data size. Therefore, we also study how diverse training samples (solid line) perform compared to the same number of randomly selected training samples (dotted line) in Fig. 9a (XSUM) and 9b (CNN). As evident, a model trained on randomly selected samples performs slightly worse on the $T_{nov}$ but obtains at least 6 (2) points higher R2 on the $T_{sim}$ subset of the XSUM (CNN) dataset for any training data size. Results further support that lexical repetitions in training reference summaries result in a significant performance gap on test samples with different levels of train lexical overlaps.

To further analyze the effect of lexical repetitions, we sort all training samples based on the maximum number of 4-gram repetitions. Then, we select the subset with the highest 4-gram repetitions such that the size of this subset is the same as the size of the subset with the least $\theta_{4G}$ ($\theta_{4G}^{nov}$)

| | CNN | XSUM | WikiHow | SamSum |
|---|---|---|---|---|
| Random | 20.14 | 19.17 | 16.72 | 26.71 |
| $\theta_{4G}^{nov}$ | 19.61 | 18.33 | 15.98 | 26.58 |
| $\theta_{sim}$ | 20.35 | 19.35 | 16.91 | 26.82 |

Table 10: Average R2 of models trained on a random, $\theta_{4G}^{nov}$ and $\theta_{sim}$ training data subsets.

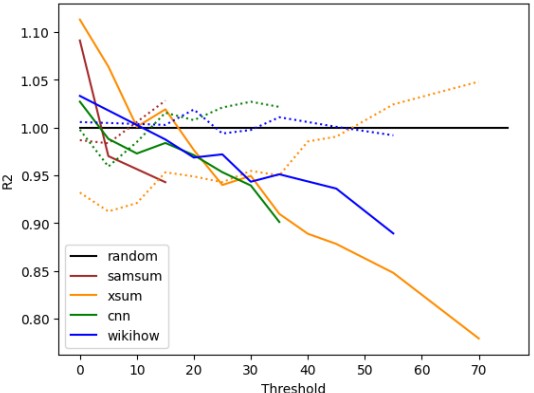

Figure 10: R2 scores comparison of *BART* models trained on a random, $\theta_{4G}^{nov}$ (solid lines) and $\theta_{sim}$ (dotted lines) training data subsets on different test data partitions.

(i.e. 2 for CNN and XSUM, 1 for SamSum, and 10 for WikiHow). We call this subset with lower diversity as $\theta_{sim}$. We also take a random subset of training data with the sample size same as the $\theta_{4G}^{nov}$. We plot the R2 of models trained with $\theta_{4G}^{nov}$ (solid lines) and $\theta_{sim}$ (dotted lines) in Fig. 10. We normalize the R2 of models trained on $\theta_{4G}^{nov}$ and $\theta_{sim}$ with the performance of the model trained on the randomly chosen subsets. As shown in Table 10, both models trained on random or $\theta_{sim}$ obtain better average R2 than the model trained on the $\theta_{4G}^{nov}$. Thus, if the expected goal from a dataset is to obtain higher average performance, controlling lexical diversity is undesirable. However, on $T_{nov}$, models trained on $\theta_{4G}^{nov}$ perform the best on all four datasets, highlighting that high lexical diversity in training summaries is important for better generalization performance of a summarization model.

## C   Intervention Analysis on XSUM

We first created a list of the 2000 most frequently mentioned entities in the training summaries, and then randomly sampled 50 entities from the list that are also present in the reference test summaries. We then edit the corresponding test document with respect to the selected entity.

If the associated fact is absent in the original source document, we explicitly add the information

| | | | |
|---|---|---|---|
| **Substituting Fact**: The Procession to Calvary, completed in `1602` -> `1802` , will remain on show at Nostell Priory in West Yorkshire. The painting had been put up for sale for £2.7m. A campaign by The Art Fund and National Trust raised £1.7m. The National Heritage Memorial Fund, which aims to save key historic items, has now stepped in with the final £1m. The painting depicts Christ carrying the cross on his way to crucifixion and has hung at Nostell Priory, a stately home near Wakefield, for 200 years. The priory is the family home of Lord St Oswald, who put it up for sale to pay for the restoration of the estate. He had said he would put it up for auction if the target was not reached by Christmas. Members of the public donated £680,000 to the campaign, with almost £510,000 coming from trusts and foundations, while The Art Fund gave a further £500,000. Art Fund director Dr Stephen Deuchar said: "Considering the economic climate, this has been a hugely challenging campaign and we are enormously grateful to all our members and supporters who have given so generously. "Working with the National Trust has been a very fruitful experience, pooling our resources to pull out all the stops and save this remarkable painting for Nostell Priory and its visitors. "Dame Jenny Abramsky, chair of the National Heritage Memorial Fund, said: "The overwhelming public support to help secure this stunning painting has been an inspiration. "Individual giving combined with ongoing support from government funds such as the National Heritage Memorial Fund will play an increasingly important role in securing our most precious heritage. "The fund's money comes from the Treasury and is intended to be the last resort for saving items of importance to the UK's national heritage. It has received £10m a year since 2007, but its grant will be halved from this year as a result of government cuts. |
| **Updated Summary**: A `incorrect: 17th Century` -> `correct: 19th century/ 200 years old /` `skipped: old/ ancient` painting of Christ carrying the cross has been saved after a campaign to buy it raised £1.8m. |
| **Adding Fact**: Michael Forney, Jonathan Boxill, James Desmarais, Matt Nickerson, David Rutherford, Mark Garside and Brandon Benedict are staying at the SSE Arena. It follows the announcement last week that captain Adam Keefe had signed a new deal to remain with the Giants for the `2021-22 season` . Belfast finished fourth in the league standings last season. |
| **Updated Summary**: Belfast Giants have announced that eight players will remain with the Elite League club for the `incorrect: 2017-18 season` -> `correct: 2021-22 season /` `skipped: next season` . |

Table 11: Examples of substituting and adding facts in a source article, and their resulting effects on generated summaries.

to the source; otherwise, we make an appropriate substitution in the source document. In both cases, if the originally generated summary contains the intervened fact, we expect a good model to replace that with the newly added/ substituted fact. Alternatively, a model skipping the intervened fact in summaries generated for either the original or intervened source article is also an acceptable behavior. On the other hand, a bad model would continue to generate the wrong fact for the intervened source article. Examples for both types of interventions and their resulting effects are shown in Table 11. We make 2 interventions for each document and evaluate three randomly trained models resulting in 300 total cases for each $\theta_{4G}$. Results are shown in Table 12.

| $\theta_{4G}$ | Correct ↑ | Skipped | Incorrect ↓ |
|---|---|---|---|
| 2 | 35.0% | 41.0% | **24.0%** |
| 5 | 35.67% | 35.67% | 28.67% |
| 10 | 40.33% | 30.67% | 29.0% |
| 25 | **45.0%** | 26.33% | 28.66% |
| all | 34.33% | 29.66% | 36.0% |

Table 12: Results for the intervention-based analysis of models trained on XSUM data with different $\theta_{4G}$. The best (worst) performing system is bolded (underlined).

We find that the model trained with the lowest $\theta_{4G}$ (fewest samples) has the highest number of

*skipped* cases. Increasing the number of training samples, with repeated 4-grams, initially increases the correctly generated facts but also makes it susceptible to making mistakes as indicated by the increased percentage of incorrect generation. The model trained on all samples makes the highest number of mistakes confirming that models are vulnerable to learning data artifacts when training summaries have many lexical repetitions. Overall, $\theta_{4G}$ of 2 makes the least number of incorrect predictions while $\theta_{4G}$ of 25 performs the best in terms of correctly updated facts according to the interventions. Consequently, when training summaries contain factual errors, we propose to control the ngram repetitions for a more accurate and consistent summarization model.