# OpenReview forum: "Lexical Repetitions Lead to Rote Learning: Unveiling the Impact of Lexical Overlap in Train and Test Reference Summaries"
_EMNLP/2023/Conference — EMNLP 2023 Findings_

### Official Review · Reviewer_Luya · 2023-07-25

**Soundness:** 4

**Excitement:**

4: Strong: This paper deepens the understanding of some phenomenon or lowers the barriers to an existing research direction.

**Paper Topic And Main Contributions:**

This paper analyzes the effect of lexical overlap in test and train reference summaries on system performance and factual errors. Generally, the experiments show that the more overlap there is, the higher the ROUGE-2 scores are (meaning that the model likely memorizes what it sees in the train set), and that models may generate more factual errors (the models use more named entities wrongly). Also, when training on subsets with different levels of overlap, more overlapping training data improves performance with similar test lexical distribution, and more overlapping training data makes models remember facts instead of inferring them from source articles. These experiments were performed on 4 different summarization datasets (with different domains and formats), and with 4 recent summarization systems. Results are somewhat different for the different datasets, depending on level of abstraction and format, but overall the results trend similarly. The resulting observations are backed by several analyses and consistent results. The overall conclusion is that there is a strong influence of the overlap between the training data and test data on the apparent performance of a model, due to misuse of related memorized n-grams in output summaries or, contrarily, over-generalizability.

**Reasons To Accept:**

- An important analysis on the behavior of models as influenced by the training and testing data used in summarization. The conclusions provide interesting insights and explanations for how to more informatively evaluate summarization tasks, and some points regarding future preparations of new datasets.
- Observations are backed by several experiments that seem sound, and with convincing explanations.
- I assume that the experimental setup can be used as an adapted framework for other tasks/datasets as well.

**Reasons To Reject:**

- It says there is a new evaluation protocol proposed, but the paper is positioned more as an analysis of many experiments, and without an explicit prioritized explanation on recommendations of what to practically do. In any case, it would nice to have simple and specific guidelines of what one should do.
- Generally, the paper was hard for me to follow, partly because (1) there were many related variables, thresholds and experiments, (2) the graphs are somewhat confusing, e.g., there are two thresholds in a single graph (maybe make graph label names more significant?), (3) the abstract is not understandable without reading the intro (maybe be more general in the abstract?), (4) not clear what "data artifacts" are.

Overall I don't see significant risks for accepting, but some improvements in presentation might help make the paper more readable.

**Reproducibility:**

3: Could reproduce the results with some difficulty. The settings of parameters are underspecified or subjectively determined; the training/evaluation data are not widely available.

**Reviewer Confidence:**

3: Pretty sure, but there's a chance I missed something. Although I have a good feel for this area in general, I did not carefully check the paper's details, e.g., the math, experimental design, or novelty.

**Typos Grammar Style And Presentation Improvements:**

- bart, brio, pegasus, etc. should be cased as in the original names, and italicized consistently throughout the paper.
- paragraph headlines are commonly not in title-case, and end with a period.
- Line 253: "of *the* two datasets"
- Line 385: what is 95-100%? Maybe be clearer about what it represents.
-Line 396: "Makes model*s*..."
- Figure 3: would be easier to read if the sub-figures' order was kept similar to the order in Figure 1
- Line 421: "editing *the* corresponding..."
- Line 423: "of *the* newly..."
- Line 463: "the entire" -> "all"
- Figure 6: mistakes in sub-figure captions (c) and (f)?
- Colors are hard to see in grayscale in the figures. But since the differences are not so important, it's not too critical.

---

> ### Author Rebuttal · Authors · 2023-08-28
>
> **Evaluation guideline**
> Thank you for the suggestion. Our proposition is to evaluate models separately on test subsets with low and high lexical overlap with training summaries. We will explicitly mention this in the future version of the paper.
>
> **Data artifacts**
> Data artifacts vary depending on the datasets. For instance, on CNN, many summaries include some journalists' names not mentioned in the source article. On XSUM, most summaries for articles related to murder include the location irrespective of whether the location information is present in the source article.
>
> Thanks for your suggestions on improving the presentation. We will incorporate them in the future version of the paper.

---

### Official Review · Reviewer_ewgz · 2023-08-04

**Soundness:** 3

**Excitement:**

2: Mediocre: This paper makes marginal contributions (vs non-contemporaneous work), so I would rather not see it in the conference.

**Paper Topic And Main Contributions:**

The paper argues that measuring the performance of a summarization model on a reference test set may be inadequate. The authors show that models tend to get a higher ROUGE-2 score for reference test summaries that have a higher lexical overlap with the train summaries. They build on this to show that such training repetitions may make the model susceptible to rote learning resulting in more hallucinations when reference test summaries are lexically close to train summaries.

**Questions For The Authors:**

R1 and R2 above.

To authors: thank you for the helpful rebuttal. I have revised my scores based on your inputs.

**Reasons To Accept:**

A1. This is an interesting insight. However, the literature seems to be moving away from reference summaries in a test set to more reference-free evals (see the G-Eval paper).  I'm convinced that this is an interesting insight that can still help us design better training sets for summarization tasks.

**Reasons To Reject:**

* R1. It is difficult to see if the arguments extend to SOTA instruction-following LLMs such as InstructGPT/Llama/PaLM2.  In some cases the training data isn't public, so hard to measure this lexical overlap.

* R2. Several open questions in the experiments:

  *    Is this true for datasets not from the news domain ?
  *    What happens if you control lexical overlap by choosing a test set from a different domain? What's the trade-off between getting a better E_Rem and  summary quality.
  *   Is this effect still true as we increase the model sizes?

**Reproducibility:**

4: Could mostly reproduce the results, but there may be some variation because of sample variance or minor variations in their interpretation of the protocol or method.

**Reviewer Confidence:**

2: Willing to defend my evaluation, but it is fairly likely that I missed some details, didn't understand some central points, or can't be sure about the novelty of the work.

---

> ### Author Rebuttal · Authors · 2023-08-28
>
> Thank you for your feedback and questions, which we answer below.
>
> **Reference-free evaluation, SoTA instruction models**
> We agree that literature is moving away from reference summaries in a test set to more reference-free evals. Also, it is unclear if the argument extends to new instruction-following LLMs, and it is difficult to measure that given the limited knowledge of their training data. However, we argue that fine-tuning summarization models on domain-specific training data is still useful for aligning models and ensure that the model summarizes only the information of interest. General LLMs such as GPT generate better human-readable summaries (https://arxiv.org/pdf/2209.12356.pdf) but their relevance/ alignment to target application (i.e. generating summaries that contain information of interest) is much worse than models fine-tuned in-domain (e.g., https://arxiv.org/pdf/2212.07981.pdf).
>
> **datasets not from the news domain**
> Yes, as shown by our experiments on WikiHow and SamSum datasets.
>
> **control lexical overlap by choosing a test set from a different domain**
> This is not the focus of our work since the test set from a new domain is expected to have lower lexical overlap with training summaries.
>
> It is worth noting that the model will remember frequent n-grams from training summaries irrespective of the lexical distribution of test summaries. Our experiments show that with more repeated training n-grams, models tend to rely less on test source articles. This is evident from our human evaluation that uses very recent news articles which are temporally exclusive from training summaries, and likely include novel summary-worthy contents. We find that higher training ngram repetitions make a model more susceptible to generating remembered facts.
>
> Anecdotally, a recent work https://aclanthology.org/2022.aacl-short.42/ also shows that summarization models tend to generate certain training n-grams even for source documents from a different domain.
>
> **trade-off between getting a better E_Rem and summary quality.**
> This is data specific, and as shown by our experiments, partly depends on the lexical repetitions in training summaries. Controlling lexical repetitions (Section 5) and performing fine-grained evaluations (Section 4)  would help in finding the right tradeoff.
>
> **increase the model sizes**
> The correlation between rote learning due to lexical repetitions and the model size is not a focus of our work. However, we used models of different sizes, for example, the pegasus model has roughly 40% more parameters than the bart-large model. We found that the rote learning phenomenon is present in both models.

---

### Official Review · Reviewer_BtBR · 2023-08-07

**Soundness:** 3

**Excitement:**

4: Strong: This paper deepens the understanding of some phenomenon or lowers the barriers to an existing research direction.

**Paper Topic And Main Contributions:**

This paper can be categorized as an NLP engineering experiment. It explores the behavior of summarization models, more precisely the effect of high lexical repetition in the training set on the factuality of generated summaries. The paper focuses on three state-of-the-art summarization models (pegasus, bart, brio) and four common summarization benchmark (CNN/DM, XSUM, WikiHow, SamSum). The authors begin by showing that all three of these models hallucinate the same span in their summaries that was not contained in the original document. Afterward, they split the datasets into slices containing a certain percentage of repeating 4-grams, and show how does the performance (ROUGE score) and factuality (entity mentions) fare - the performance drops as the dataset is narrowed down, but the factuality (entity overlap) at some point improves. Later, they conduct experiments with fine-tuning and calibrating models on dataset subsets with particular repeating 4-gram percentages, and once again confirm their hypotheses. They end with a quick user study and conclusion.

**Questions For The Authors:**

A: In line 241, summaries and source articles are labeled with "non-hallucinated" in the parentheses. What does this refer to? If there are made-up entities in a generated summary, isn't it then hallucinated?
B: In the user study, what exactly is "the most relevant information for each document", is it the entities? Were the annotators given guidelines to align? How were the summaries compared summaries for relevance and consistency?

**Reasons To Accept:**

The paper investigates an important problem in all modern generative models, namely the hallucinations and factuality of generated text. It provides nice findings regarding the origin of hallucinations (repeating n-grams in training datasets) and proposes a practical solution on how to tackle the problem (fine-tuning + calibrating). There is a wide array of experiments performed and lots of valuable results reported in the paper. The authors also conduct a user study, which is always important in NLG tasks.

**Reasons To Reject:**

While the research problem is interesting and results extensive, paper could have been written with more clarity. Some of the technical details in the implementation are not properly explained or just assumed to be well-known to the readers. For metrics such as entity recall, entity precision, and remembered entities, how exactly are these entities detected (some tool or method), what type of entities are they (named, temporal, numerical...), what are the formulas used. In general, the paper should be more self-contained in the main 8 pages, since currently the results for 2 out of 4 datasets (WikiHow and SamSum) are only described in the Appendix - they could be briefly described in the main part. Lastly, the user study is also not fully described and lacks details in methodology and execution.

**Reproducibility:**

3: Could reproduce the results with some difficulty. The settings of parameters are underspecified or subjectively determined; the training/evaluation data are not widely available.

**Reviewer Confidence:**

4: Quite sure. I tried to check the important points carefully. It's unlikely, though conceivable, that I missed something that should affect my ratings.

---

> ### Author Rebuttal · Authors · 2023-08-28
>
> **technical details in the implementation**
> Thanks for pointing this out, we will include these details in the future version. We used Spacy (en_core_web_sm) for identifying entities and used token-level overlap for calculating different metrics. Since a model may remember/ hallucinate any type of entity, we use all entity types (excluding stop words) in our metrics.
>
> **paper should be more self-contained in the main 8 pages**
> It was difficult to include all the results in eight pages. So, to at least briefly discuss each experiment/ evaluation, we chose to include all results for CNN and XSUM datasets in the main paper. Given an additional page in the final version, we plan to briefly discuss findings on WikiHow and SamSum datasets in the main paper, while leaving detailed results for the appendix.
>
> **In line 241, summaries and source articles**
> Yes, made-up entities in generated summaries are hallucinations. Entity recall measures the percentage of generated entities that are present in both the source article and reference summaries. These generated entities are not hallucinations, that’s why we also included non-hallucinated in the parentheses. We will rephrase the definition of entity recall, and include the equation to avoid any confusion.
>
> **Human Study**
> In human study, relevance is not limited to entities. First, annotators highlighted 2-5 most relevant information (which includes events and entities) in a document. Then, we compare the number of document highlights that are present in summaries generated by the two systems. If both systems include the same number of highlights, both systems are labeled equivalent. Otherwise, the system that generated more highlighted information is rated better. For consistency, we compare the number of hallucinated facts in summaries generated by two systems and rank the system with fewer hallucinations as better. We will include these details in the future version of the paper.

---

### Meta-Review · Area_Chair_5y41 · 2023-09-19

**Recommendation:** 4

**Metareview:**

The paper analyzes the problem of the presence of words or n-grams contained in the references of the training set in the automatically generated summaries. Four different corpora for summarization and several models for automatic summarization are used in the study. Different conclusions are presented, which are interesting for the improvement of the results in this area. Even with the widespread use of pre-trained language models to generate summaries, it still seems to be necessary to have corpora such as the ones analyzed for fine tuning and improving the summaries generated.

---

### Decision · Program_Chairs · 2023-10-07

**Decision:**

Accept-Findings

**Comment:**

The paper analyzes the problem of the presence of words or n-grams contained in the references of the training set in the automatically generated summaries. Four different corpora for summarization and several models for automatic summarization are used in the study. Different conclusions are presented, which are interesting for the improvement of the results in this area. Even with the widespread use of pre-trained language models to generate summaries, it still seems to be necessary to have corpora such as the ones analyzed for fine tuning and improving the summaries generated.